# Genome-Wide Analysis of the Amino Acid Permeases Gene Family in Wheat and *TaAAP1* Enhanced Salt Tolerance by Accumulating Ethylene

**DOI:** 10.3390/ijms241813800

**Published:** 2023-09-07

**Authors:** Kai Wang, Mingjuan Zhai, Dezhou Cui, Ran Han, Xiaolu Wang, Wenjing Xu, Guang Qi, Xiaoxue Zeng, Yamei Zhuang, Cheng Liu

**Affiliations:** 1Crop Research Institute, Shandong Academy of Agricultural Sciences/National Engineering Research Center of Wheat and Maize/Shandong Technology Innovation Center of Wheat, Jinan 252100, China; 18211084814@163.com (K.W.); dezhoucui@126.com (D.C.); hr022cn@aliyun.com (R.H.); xiaoluwang1989@hotmail.com (X.W.); wenjingxu1989@163.com (W.X.); guangq1102@163.com (G.Q.); zengxiaoxue@hotmail.com (X.Z.); 18954218411@163.com (Y.Z.); 2Biotechnology Research Institute, Chinese Academy of Agricultural Sciences, Beijing 100081, China; 15010803974@163.com

**Keywords:** amino acid permeases, gene family, salt stress, ethylene, *Triticum aestivum*

## Abstract

Amino acid permeases (AAPs) are proteins of the integral membrane that play important roles in plant growth, development, and responses to various stresses. The molecular functions of several AAPs were characterized in *Arabidopsis* and rice, but there is still limited information on wheat. Here, we identified 51 *AAP* genes (*TaAAPs*) in the wheat genome, classified into six groups based on phylogenetic and protein structures. The chromosome location and gene duplication analysis showed that gene duplication events played a crucial role in the expansion of the *TaAAPs* gene family. Collinearity relationship analysis revealed several orthologous *AAPs* between wheat and other species. Moreover, cis-element analysis of promoter regions and transcriptome data suggested that the *TaAAPs* can respond to salt stress. A *TaAAP1* gene was selected and transformed in wheat. Overexpressing *TaAAP1* enhanced salt tolerance by increasing the expression of ethylene synthesis genes (*TaACS6*/*TaACS7*/*TaACS8*) and accumulating more ethylene. The present study provides an overview of the *AAP* family in the wheat genome as well as information on systematics, phylogenetics, and gene duplication, and shows that overexpressing *TaAAP1* enhances salt tolerance by regulating ethylene production. These results serve as a theoretical foundation for further functional studies on *TaAAPs* in the future.

## 1. Introduction

Amino acid transporters (AATs) are integral membrane proteins that are crucial for plant growth and development, including long-distance amino acid transport and responding to pathogen and abiotic stress in higher plants [1,2,3]. In plants, AATs can be classified into two main families: the amino acid/auxin permease (AAAP) and the amino acid, polyamine, and choline (APC) families. Of these, the AAAP family can be further divided into six subfamilies: amino acid permeases (AAPs), lysine and histidine transporters (LHTs), proline transporters (ProTs), c-amino butyricacid transporters (GATs), auxin transporters (AUXs), and aromatic and neutral amino acid transporters (ANTs). Meanwhile, the APC family can be grouped into three subfamilies: cationic amino acid transporters (CATs), amino acid/choline transporters (ACTs), and polyamine H^+^-symporters (PHSs) [4,5]. Hence, AAPs are a plant-specific family and were first discovered in *Arabidopsis* by complementation of yeast strains deficient in amino acid uptake [6,7,8].

At present, the functions of several AAPs have been characterized in plants. Eight members of the *AtAAP* genes family (AtAAP1-AtAAP8) exist in *Arabidopsis*, and several of them are involved in the uptake and transport of amino acids from roots to embryo or phloem loading, including *AtAAP1*/*2*/*5*/*6*/*8* [9,10,11,12,13,14,15]. The AAPs gene family consists of 19 members in rice (*OsAAP1*-*19*). *OsAAP6* functions as a positive regulator of grain protein content and its overexpression greatly enhances root absorption of a range of amino acids and influences the distribution of various amino acids. However, this gene has no known effects on plant morphology, flowering time, and grain yield, whereby it can be used for the genetic improvement of grain protein content and grain nutritional quality in rice [16]. *OsAAP1* is highly expressed in rice axillary buds, leaves, and young panicles, and functions as a positive regulator of growth and grain yield in rice by increasing neutral amino acid uptake and reallocation [17]. *OsAAP3* is ubiquitously expressed across all plant organs, especially roots and inflorescences, and is mainly responsible for transporting Lys and Arg from the intercellular space to the plant cells. The inhibition of *OsAAP3* expression increases grain yield due to the formation of large numbers of tillers as a result of increased bud outgrowth [18,19]. *OsAAP7* is highly expressed in roots and leaves, while *OsAAP16* has high expression levels in roots, leaves, flowers, and seeds. OsAAP7’s and OsAAP16’s substrate specificity is similar to that of OsAAP1, with the former proteins transporting greater amounts of arginine than the latter [20]. Furthermore, the functions of AAPs have also been studied in other plant species, with potato *StAAP1* involved in sink-to-source transition and free amino acid content [21,22]. Moreover, the overexpression of *Vicia faba VfAAP1* in the cotyledon parenchyma of *Faba bean* and *Vicia narbonensis* results in an increase in seed sink strength for nitrogen, improves plant nitrogen status, and leads to higher seed protein, weight, and size [23]. Recently, PtAAP11 has been proposed as a high-affinity amino acid transporter that plays a major role in xylogenesis by providing proline in poplar plants [24].

Salt stress is one of the major abiotic stresses, making it a major threat to crop productivity. Salt stress negatively influences seed germination, growth, and reproduction, and sometimes even results in death under severe conditions. Salinity stress decreases the capacity for water absorption by the roots and increases the transpiration rate due to osmotic imbalance, resulting in closed stomata and reducing the photosynthesis rate and carbohydrate production [25]. Salt stress also accumulates reactive oxygen species (ROS), leading to the uncontrolled oxidation of membranes, proteins, and DNA, ultimately resulting in cell death [26]. To deal with salt stress, salt-adapted plants have evolved complex mechanisms to tolerate salt stress, including physiological and biochemical processes, and developmental and morphological changes [27,28]. Ethylene, the first gaseous plant hormone to be identified, is a key regulator of salt stress tolerance in plants. Earlier work showed that ethylene biosynthesis and signaling are implicated in salinity stress tolerance in plants. Overproduction of endogenous ethylene or exogenous treatment of ethylene precursors such as 1-aminocyclopropane-1-carboxylic acid (ACC) increase salt stress tolerance in *Arabidopsis* and maize [29], suggesting a positive regulatory role of ethylene in salt-tolerance stress. However, negative regulation of ethylene in salt stress has also been reported in rice. Exogenous treatment with ethylene in rice resulted in salt hypersensitivity [30]. Elevation of ethylene production under salinity stress significantly reduced growth, grain filling, and development of spikelets in rice [31]. Exogenous application of 1-Methylcyclopropene (1-MCP), an ethylene action inhibitor, to the rice spikelets resulted in improved physiological, agronomical, and biochemical characteristics under salinity stress, suggesting a negative role of ethylene in salt stress tolerance in rice [32]. In general, research carried out to date on ethylene and salt stress confirms that ethylene levels may positively or negatively affect plants’ responses to salt stress.

Wheat (*Triticum aestivum*) is one of the most important staple foods and plays a crucial role in sustaining food security. However, its production is greatly threatened by abiotic stresses such as salt [33]. To accommodate the stress conditions, wheat has evolved elaborate defense mechanisms to enable survival under environmental stress [34]. Therefore, discovering the genes involved in salt stress tolerance and cultivating genetically modified plant varieties with enhanced stress tolerance are currently among the most important goals for plant breeders [35]. Despite the aforementioned advances, rare TaAAPs have so far only been functionally characterized in wheat, especially in response to salt stress. This study provides comprehensive analyses of *TaAAPs*, including the identification of 51 *TaAAPs* family members, the establishment of phylogenetic relationships between *TaAAPs* and *AAPs* of other species, the evaluation of the genetic structure, conservation, and chromosomal distribution of *TaAAPs*, and the expression profiling of *TaAAPs* in salt stress using available transcriptomic data. Importantly, we found *TaAAP1* was highly decreased upon salt stress, and overexpression of *TaAAP1* increased the germination energy of seeds and seedling tolerance to salt stress by accumulating more ethylene under salt stress. Our results will benefit the functional validation of *TaAAPs* genes and broaden our understanding of their roles in wheat plants.

## 2. Results

### 2.1. Identification of the Amino Acid Permease Gene Family in Wheat

A total of eight amino acid permeases were found in *Arabidopsis*, all of which contain the conserved amino acid_trans domain (Pfam: PF01490). We identified 217 coding sequences corresponding to PF01490 in the wheat genome using the WheatOmics database (http://202.194.139.32/tools/proteinfamily.html, accessed on 13 December 2022) with PF01490 as a query. These sequences included 116 amino acid transporters, seventy amino acid permeases, fifteen auxin influx transporters, nine proline transporters, six lysine/histidine transporters, and one histidine amino acid transporter (Appendix A). We then downloaded sequences of 70 amino acid permeases and confirmed the presence of the conserved domain using Batch CD search tools (https://www.ncbi.nlm.nih.gov/Structure/bwrpsb/bwrpsb.cgi, accessed on 13 December 2022). We found that a total of fifty-one proteins contain the conserved Aa_trans domain, thirteen proteins contain the SLC5-6-like_sbd domain, and six proteins possess RHOD, STKc_IRAK, PLN02878, F-box_SF, NADB_rossmann, and DUF4220 domains, respectively (Appendix A). These 51 proteins were selected for further research and termed TaAAP1-TaAAP51. The abundance of amino acid permeases indicates their diverse functional roles in plant growth and development.

We then predicted gene characteristics, protein sequences, amino acid numbers, protein molecular weight (WM), theoretical pI (isoelectric point) (https://web.expasy.org/compute_pi/, accessed on 14 December 2022), subcellular localization (http://www.csbio.sjtu.edu.cn/bioinf/plant-multi/, accessed on 14 December 2022), and transmembrane numbers (https://services.healthtech.dtu.dk/services/TMHMM-2.0/, accessed on 14 December 2022) (Appendix A). TaAAP51 (TraesCS7D02G388300) was the largest of the 51 proteins with 527 amino acids, while TaAAP42 (TraesCS7A02G194500) was the smallest with 420 amino acids. The range of MW and pI were 45.8 to 57.5 KDa and 7.63 to 9.24, respectively. All AAPs contained several transmembrane domains, ranging from seven to eleven. The subcellular localization results showed that thirty-four AAPs were located in the cell membrane, five AAPs in the Golgi apparatus, and twelve AAPs in both the cell membrane and the Golgi apparatus (Appendix A). This suggests that TaAAPs participate in amino acid transport and protein processing.

### 2.2. Phylogenetic Analysis of AAPs in Wheat, Rice, and Arabidopsis

To better understand the phylogenetic relationship of AAPs between wheat, *Arabidopsis*, and rice, we constructed an unrooted phylogenetic tree with the neighbor-joining method using MEGA 6.0 (Figure 1). The tree consisted of fifty-one wheat, nineteen rice, and eight *Arabidopsis* sequences that could be divided into six groups according to their phylogenetic relationships: Group I contained eight, five, and three; Group II contained eleven, three, and zero; Group III contained seven, three, and one; Group IV contained five, one, and zero; Group V contained nine, four, and zero; Group VI contained twelve, three, and four AAPs from wheat, rice, and *Arabidopsis*, respectively. The five AAPs of Group IV are not present in *Arabidopsis* or rice; whereby different AAP’s transport activity or function may remain unknown in these species.

### 2.3. Phylogenetic, Gene Structure, and Conserved Motif Analyses of Wheat AAPs

We next performed phylogenetic, gene structure, and conserved motif analyses to further understand phylogenetic relationships between *AAPs*. Conserved motif analysis of TaAAPs was performed using protein sequences with the MEME online tool. We analyzed a total of 10 motifs in each gene and termed them accordingly, from 1 to 10. We found that 39 TaAAPs contain 10 motifs, some of which (e.g., motif2, motif5, motif6, and motif10) were distributed across the remaining 12 TaAAPs. The presence of these motifs in all genes indicates they are necessary for basic functions (Figure 2A). We then used the coding sequence (CDS) and genomic sequence information of corresponding genes to perform gene structure analysis using the GSDS online tool. We found significant differences in exon–intron distribution, implying these genes might have different functions and underwent distinct evolutionary processes (Figure 2B). In contrast, similar motifs and intron/exon structures indicate similar functions. Overall, phylogenetic, conserved motif, and gene structure analyses of *TaAAPs* confirmed the reliability of group classifications in wheat.

### 2.4. Chromosomal Location and Collinearity Relationship Analyses of AAPs in Wheat, Rice, and Arabidopsis

We found *TaAAP* members are distributed heterogeneously across the 42 wheat chromosomes (Figure 3), with five *TaAAPs* in chromosomes 2B, 2D, 3B, and 7D; four *TaAAPs* in 2A, 3A, and 3D; three *TaAAPs* in 7A and 7B; two *TaAAPs* in 5A, 5B, and 5D; and one *TaAAP* in 1A, 1B, 1D, 4A, 4B, 4D, and 6A. Accordingly, the number of *TaAAP* genes in the A, B, and D sub-genomes was 16, 17, and 18, respectively. A previous study defined a tandem duplication event as cases when there are two or more duplicated genes within 200 kb [36,37]. Here, we found five gene pairs (*TaAAP19*/*20*/*21*, *TaAAP23*/*23*, *TaAAP24*/*25*/*26*, *TaAAP27*/*28*, and *TaAAP39*/*30*) arranged in tandem that documented the expansion of the *TaAAP* family during the evolutionary process (23.6%, 12 of 51). These tandem duplications mainly occurred on Chromosomes 3A/3B/3D (Figure 3).

To analyze orthologous relationships and the origin of *AAPs* in different species, the molecular histories of AAP family members were further analyzed. We found a highly conserved collinear relationship between wheat, rice, and *H. vulgare* (Figure 4), with 26 wheat genes having a syntenic relationship with *O. sativa* and *H. vulgare*. The number of orthologous AAP pairs in wheat and other species (*O. sativa* and *H. vulgare*) were 37 and 38, respectively. Some AAP family members were associated with more than one syntenic gene pair in wheat and rice, or *H. vulgare* (Figure 4). For example, *TaAAP1* was associated with *OsAAPs* in chromosomes 1 and 5 (*OsAPP7*) and was associated with *HvAAPs* in chromosomes 1H and 3H, implying these genes have the same ancestor and are functionally conserved. We also found no collinearity between *Arabidopsis* and wheat or rice (Appendix A). Overall, our results suggest these orthologous gene pairs evolved after the divergence between monocots and dicots.

### 2.5. cis-Regulatory Elements in Wheat TaAAPs

To further investigate the regulatory mechanisms associated with the *TaAAPs*, we exacted cis-regulatory elements in the promoter regions of *TaAAPs*. Specifically, a 2000 bp sequence located upstream of the translational start site was considered as the putative promoter region and thus analyzed for the presence of cis-regulatory elements. We found multiple cis-regulatory elements involved in jasmonic acid (JA) responsiveness (the CGTCA motif and the TGACG motif) and salicylic acid (SA) (TCA element) in the *TaAAP* promoter regions. For example, *TaAAP1*/*2*/*3*/*4* contain multiple TCA motifs and TC-rich repeats, and *TaAAP5*/*13*/*14*/*20*/*35*/*36* contain multiple CGTCA motifs, implying that these genes might participate in pathogen response. In addition, multiple elements involved in abscisic acid (ABA) responsiveness (ABRE), drought (MBS), and low temperature responsiveness (LTR) were also detected (e.g., *TaAAP14*/*35*/*36*/*38*), and may be involved in responses to biotic stress. Interestingly, we found some *TaAAPs* promoter contain cis-regulatory elements involved in seed and endosperm expression (Figure 5), highlighting the diverse modes of action of these genes.

### 2.6. TaAAP1 Overexpression Wheat Improves Salt Tolerance in Wheat

To assess the roles of *TaAAPs* in salt stress response, we treated Jimai 22, the modern winter wheat variety, with 200 mM NaCl for 0/2/12/48 h and collected seedlings for RNA sequencing. The data showed that the expression of *TaAAP1/2/3*/*4*/*8*/*13* decreased after NaCl treatment, while that of *TaAAP5*/*9*/*14*/*16*/*25*/*31*/*32*/*33*/*35* increased, implying these genes may involve in salt stress (Appendix A, Appendix A). Considering that *TaAAP1/2/3*, *TaAAP4/8/13*, *TaAAP5/9/14*, and *TaAAP31/32/33* are homologous, respectively, we hypothesize that these genes are salt stress-responsive genes in wheat. Fortunately, we collected lines overexpressing *TaAAP1*. The coding sequence of *TaAAP1* from Jimai 22 was cloned and introduced into the hexaploid common wheat cultivar ‘Fielder’. Two overexpressed lines (OE1/OE2) were selected to test the responses of *TaAAP1* to salt stress (Appendix A).

The germination and seedling stages are key periods for salt sensitivity in wheat. We first calculated the germination rate of Fielder and overexpression lines under normal and 200 mM NaCl conditions. The results showed no difference in germination rate between the transgenic and Fielder lines under normal conditions and 200 mM NaCl, but the germination energy was significant higher in the overexpression lines than in Fielder (Figure 6A). This means that *TaAAP1* overexpression improved germination under salt stress. Then, we tested the responses of seedlings under normal and 200 mM NaCl conditions. Seven-day-old seedlings were planted in 1/2 Hoagland solution with or without 200 mM NaCl, respectively, for 10 days to record the growth phenotype. Results showed no differences in shoot length and root length between the overexpression and Fielder lines under normal conditions, but significant inhibition of root length in both transgenic and Fielder lines under salt stress. In addition, the transgenic lines had a longer root than the Fielder lines (Figure 6B), meaning that *TaAAP1* overexpression improved salt tolerance.

### 2.7. TaAAP1 Overexpression Wheat Accumulated More Ethylene under Salt Stress

Ethylene, a gas phytohormone, plays a significant role in plant responses to salt stresses. We examined ethylene production in transgenic plants and Fielder plants under normal and 200 mM NaCl conditions. We firstly tested the expression levels of *TaAAP1* in transgenic lines under normal and salt stress, results showed that *TaAAP1* was highly expressed in transgenic wheat compared with Fielder under normal condition and salt stress (Figure 7A), providing that *TaAAP1*-overexpressing lines were valid. We then measured the ethylene content and found that the ethylene content in Fielder plants was 15.6 μg/L/24 h, while the values in OE1 and OE2 were 39.6 and 46.2 μg/L/24 h, respectively, under normal conditions (Figure 7B). After salt stress, ethylene accumulation was significantly higher in both overexpression and Fielder lines. Ethylene production in Fielder was 41.4 μg/L/24 h, while the values in OE1 and OE2 were 54.6 and 68.1 μg/L/24 h, respectively (Figure 7B). We then tested the expression levels of the expression of *ACS2*, *ACS6*, *ACS7*, *ACS8*, and *ACO1* as these are key rate-limiting enzymes that catalyze the committing reaction in ethylene biosynthesis. RT-qPCR showed that *TaAAP1* overexpression increased expression of *TaACS6*/*TaACS7*/*TaACS8* in both normal conditions and salt stress, and increased expression of *TaACO1* only in normal conditions, with no effect on *TaACS2* expression (Figure 7C). This result showed that salt stress induces ethylene production and *TaAAP1* transgenic plants accumulated high levels of ethylene under normal and salt stress conditions by increasing the expression of ethylene biosynthesis genes.

To further validate the roles of ethylene in wheat responses to salt stress, we treated Fielder seeds under normal and 200 mM NaCl conditions with 0 or 50 μg/mL ethephon and calculated the germination rate. The results showed that the germination rate was the same in Fielder lines under normal conditions with 0 or 50 μg/mL ethephon, but the germination rate was significant higher in 50 μg/mL ethephon than in 0 under 200 mM NaCl conditions (Figure 7D). This result implies that ethylene might play a positive role in wheat responses to salt stress, at least for seed germination.

## 3. Discussion

### 3.1. Organization of AAP Family Genes in Wheat

Previous studies identified and functionally characterized eight and nineteen *AAPs* in *Arabidopsis* and rice, respectively. However, members of the AAP family in wheat remain unknown. In this study, we identified 51 *TaAAP* genes distributed across six groups based on amino acid sequence similarity. In *Arabidopsis*, eight AtAAPs were divided into three groups, containing AtAAP1/6/8 (Group I), AtAAP2/3/4/5 (Group VI), and AtAAP7 (Group III) (Figure 1). We also note that the numbers of TaAAPs are significantly larger in wheat (51 in total) compared to *Arabidopsis* and rice, owing to differences in genome size (17 Gb for wheat, 125 Mb for *Arabidopsis*, and 466 Mb for rice). The increase in the number of *TaAAPs* also suggests successful family expansion and rearrangement, with these transporters playing an important role in adaptation to specific functions.

Chromosomal mapping of *TaAAP* genes showed variable distribution across 39 wheat chromosomes, with most members localized on chromosomes 2B, 2D, 3B, and 7D. We also found tandem duplications on chromosomes 3A, 3B, 3D, and 7D (Figure 3). Gene structure and conserved motif analysis revealed most genes in the same group are structurally conserved in the number of introns and exons, as well as conserved motifs (Figure 4), which indicate close evolutionary relationships.

### 3.2. TaAAP1 Is a Positive Regulator in Wheat Response to Salt Stress

The role of *AAPs* in amino acid uptake and transport has been extensively studied in *Arabidopsis* and rice, but their role in the response to biotic/abiotic stress is limited. For example, the expression of *AtAAP4* and *AtAAP6* is downregulated by salt stress in *Arabidopsis*, while *OsAAP4*/*6*/*8*/*11* are regulated by salt and drought stresses. Based on the expression profile of wheat treated with salinity (Appendix A), we demonstrated that several *TaAAPs* respond to salt stress. Fortunately, we have applied the *TaAAP1*-overexpression lines from Doctor Han. A salt-response experimental assay showed that *TaAAP1* overexpression enhances salt tolerance at the germination and seedling stages, as these two stages are key periods for salt sensitivity in wheat. Seed germination is a critical stage that initiates the life cycle of a plant and is severely affected by salt stress (Figure 6A,B). At the germination stage, the germination energy is significant higher in *TaAAP1*-overexpression lines than Fielder lines under salt stress, implying that *TaAAP1* overexpression boosted germination and root growth to absorb water and balance osmotic stress. After direct seeding in soil, the emerging primary root must quickly adapt to rapidly changing environmental conditions, including salt. Primary root vigor is an important selection criterion for salt-tolerant plants. At the seedling stage, the seedlings of transgenic lines have longer roots than Fielder seedlings under salt stress, proving that *TaAAP1* overexpression led to longer roots, which is beneficial to maintaining osmotic stress and ionic homeostasis during salt stress. The U-box E3 ubiquitin ligase TaPUB1 positively regulates salt stress tolerance in wheat. The *TaPUB1*-OE transgenic plants had longer roots and shoots, better growth status, and higher photosynthetic capacity than WT plants under salt stress, but these phenotypic characteristics were the opposite in *TaPUB*-RNAi plants [34].

### 3.3. TaAAP1 Overexpression Wheat Accumulates Ethylene

Ethylene is a gaseous hormone that regulates plant growth and development. Increasing evidence suggests that ethylene is also a stress hormone because its synthesis is induced by various biotic and abiotic environmental stresses [38]. The roles of ethylene in salt stress responses remain an open question. In *Arabidopsis*, salinity promotes ethylene accumulation by modulating the activity of enzymes regulating ethylene biosynthesis (e.g., ACS2 and ACS7) [39,40]. Overproduction of endogenous ethylene or treatment with the ethylene precursor ACC can overcome the salt-induced restraint of *Arabidopsis* seed germination and promote the salinity tolerance of seedlings grown on saline soil at the vegetative growth stage [41,42]. In rice, SIT1, a lectin receptor-like kinase, positively regulates salinity tolerance by promoting the activity of the MAPK3/6 protein kinase and promoting ethylene production [43]. However, in some other cases, elevated ethylene levels can adversely affect salinity tolerance. For instance, *Arabidopsis* plants overexpressing *TaACO1* display elevated ethylene levels and decreased salinity tolerance [44]. The *Arabidopsis acs7* mutant reduces ethylene production and exhibits increased salt tolerance at the seed germination stage [40]. Exogenous treatment with ethylene in rice resulted in salinity hypersensitivity, while exogenous application of 1-MCP to the rice spikelets resulted in improved physiological, agronomical, and biochemical characteristics under salinity stress [32]. Taken together, these various studies indicate that ethylene can either negatively or positively affect the salinity sensitivity of plants, suggesting that the fine-tuning of ethylene biosynthesis might be essential to salinity tolerance in plants. Therefore, we found that salt stress can lead to ethylene accumulation. In addition, *TaAAP1* overexpression boosted ethylene production (Figure 7B), implying that *TaAAP1* overexpression enhanced salt tolerance by elevating ethylene content in salt stress. The RT-qPCR assay proved that *TaAAP1* can upregulate the expression of several ethylene biosynthesis genes, including *TaACS6*/*TaACS7*/*TaACS8* (Figure 7C). Exogenous application of ethephon promotes seed germination under salt stress (Figure 7D), implying that ethylene might be a positive regulator in the response of wheat to salt stress. The function of ethylene needs to be confirmed through mutants. Future studies are needed to investigate which amino acids are specifically transported by TaAAP1 and how TaAAP1 influences the expression of ethylene biosynthesis genes. Such studies will help identify vital genes responding to salt stress, and contribute to a better understanding of *TaAAP* genes functions with respect to abiotic stress.

## 4. Materials and Methods

### 4.1. Sequence Identification and Annotation of TaAAP Genes

We downloaded genome files, protein sequences, and GFF3 (General Feature Format Version 3) files for *Triticum_aestivum*, *Arabidopsis_thaliana*, *Oryza_satival*, and *Hordeum_vulgare* from the ensembl plant website (http://plants.ensembl.org/index.html, accessed on 10 December 2022). The specific domain of amino acid permease (PF01490) was downloaded from the Pfam database (http: //pfam.sanger.ac.uk/, accessed on 13 December 2022). The WheatOmics website (http://202.194.139.32/tools/proteinfamily.html, accessed on 13 December 2022) was used to screen wheat amino acid permeases (AAPs) based on PF01490, which allowed us to originally identify 217 candidates that were further identified using the NCBI Conserved Domain database (https://www.ncbi.nlm.nih.gov/Structure/bwrpsb/bwrpsb.cgi, accessed on 13 December 2022). This resulted in 51 verified TaAAPs. The amino acid sequences, molecular weights (MW), and isoelectric points (pI) of the identified TaAAPs were predicted using the ExPasy website (https://www.expasy.org/, accessed on 14 December 2022). The subcellular localization predictions of TaAAPs members were performed using the Plant-mPLoc website (http://www.csbio.sjtu.edu.cn/bioinf/plant-multi/, accessed on 14 December 2022) [45]. The transmembrane numbers were predicted using the TMHMM-2.0 website (https://services.healthtech.dtu.dk/service.php?TMHMM-2.0, accessed on 14 December 2022).

### 4.2. Phylogenetic Analysis of AAPs Genes

*Arabidopsis*, rice, and wheat AAPs protein sequences were downloaded and compared to further clarify evolutionary relationships. Full-length protein alignments were performed using MUSCLE 3, and a phylogenetic tree was constructed using the neighbor-joining method with 1000 bootstrap replications with MEGA-6 (https://www.megasoftware.net/, accessed on 13 February 2023) [46].

### 4.3. Conserved Domains, Motifs, and Gene Structures of TaAAP Genes

The conserved domains present in TaAAPs sequences were identified using the NCBI Conserved Domain Search (CD Search) tool (https://www.ncbi.nlm.nih.gov/Structure/bwrpsb/bwrpsb.cgi, accessed on 15 February 2023). The intron-exon structure of *TaAAP* genes was examined using the GSDS 2.0 (http://gsds.gao-lab.org accessed on 15 February 2023) online tool [47]. The TaAAPs conserved domains were visualized using TBtools [48], and the conserved protein motifs identified using MEME (https://meme-suite.org/meme accessed on 15 February 2023). The parameters were set to an expectation of 10 motifs distributed in the sequences, with no repetitions allowed.

### 4.4. Chromosomal Distribution and Synteny Analysis of TaAAPs

By analyzing the wheat genome annotation, we mapped *TaAAPs* to their respective chromosomal locations and obtained a gene density profile for each chromosome. For the synteny analysis, we used the one-step MCScanX to evaluate gene collinearity events. Subsequently, we utilized TBtools v2.0 to visualize chromosomal localizations and gene synteny in different species [48].

### 4.5. Cis-Elements and Expression Analysis of TaAAPs

To further identify putative cis-regulatory elements in the promoter regions of *TaAAPs*, we obtained DNA sequences 2 kb upstream *TaAAP* genes using TBtools. The various putative cis-regulatory elements of these sequences were further analyzed using PlantCARE databases (http://bioinformatics.psb.ugent.be/webtools/plantcare/html, accessed on 16 February 2023) [49]. The diagram was visualized using TBtools [48].

### 4.6. Expression Analysis of the TaAAP Gene Family from RNA-Seq Data

To analyze the expression patterns of *TaAAP* genes in salt stress, we collected transcriptomic data of Jimai 22 seedlings with or without 200 mM NaCl treatment from Dezhou Cui. The expression levels of *TaAAPs* were then calculated and heat maps were generated using TBtools.

### 4.7. Salt and Ethephon Treatment

The wide-type wheat Fielder and *TaAAP1*-overexpressing lines were provided by Ran Han (Crop Research Institute, Shandong Academy of Agricultural Sciences), and T3 homozygous transgenic lines were used to perform the salt stress treatment.

For the germination assay, fifty seeds of Fielder and *TaAAP1*-overexpressing lines were germinated on filter paper moistened with sterile distilled water or 200 mM NaCl. The germination rate was calculated every day, with three biological replicates.

For the seedling assay, ten germinated wheat seedlings (7 days old) with coincident growth were selected and transplanted into a rectangle box (1 L) containing 1/2 Hoagland solution with or without 200 mM NaCl, respectively. Subsequently, they were grown at 22 °C with 16 h light/18 °C with 8 h dark for 10 days. Three biological replicates were used to measure the root and shoot length. The phenotypes of the wheat seedlings were photographed.

For ethephon treatment, one hundred seeds of Fielder were germinated on filter paper moistened with 200 mM NaCl with or without 50 μg/mL ethephon. The germination rate was calculated every day, using three biological replicates.

### 4.8. Measurement of Ethylene Content

Ethylene emission was measured with a gas chromatograph as described previously [50]. In brief, fifteen germinated seedlings (2 days old) with coincident growth were selected and grown in a 175 mL conical flask containing filter paper and moistened with 1/2 Hoagland solution for one week. Seedlings were treated with 1/2 Hoagland solution with or without 200 mM NaCl and sealed tightly for 24 h. In total, 1 mL of gas from each vial was used to analyze the ethylene emissions with a gas chromatograph TRACE 1300 (Thermo Fisher Scientific S.p.A, Milan, Italy) by Nanjing Convinced-Test Technology Co., Ltd. (Nanjing, China).

### 4.9. Quantitative Real-Time PCR

Total RNA was extracted from wheat using Trizol reagent (Thermo Fisher Scientific Life Sciences Solutions, CA, USA). RNA was first purified and then checked for integrity. For RT-qPCR, total RNA was reverse-transcribed to cDNA using the FastQuant RT Kit (Tiangen Biotech Co., Ltd., Beijing, China). RT-qPCR was carried out in a Roche LightCycler480 system using the SYBR Premix Ex Taq kit (Takara Biomedical Technology (Beijing) Co., Ltd, Beijing, China). Reactions were set up using the following thermal cycling profile: 95 °C for 30 s followed by 40 cycles of 95 °C for 5 s, 58 °C for 30 s, and 72 °C for 34 s. Each experiment was replicated three times. The relative expression of the target genes was calculated using the 2^−∆∆CT^ method, with the wheat Actin gene (*TaActin*) used as the internal reference gene for the analysis. Primers are listed in Appendix A.

### 4.10. Construction of TaAAP1 Overexpression Lines

To obtain *TaAAP1*-OE plants, the coding sequence of *TaAAP1* was cloned from Jimai 22 and subcloned into the wheat pLGY-OE3 vector with the Ubi promoter. The vectors containing the target gene were introduced into agrobacterium tumefaciens strain EHA105 and transformed by agrobacterium-mediated infection into immature embryos of the hexaploid wheat variety Fielder. Positively transgenic plants were selected based on the resistance to bar and by PCR detection. Two independent lines in T2 with high expression levels were used in this study.

### 4.11. Statistical Analyses

All experimental results are means ± standard deviations of at least three independent replicates. Student’s *t*-test was used for comparing two data sets in the SPSS System (IBM SPSS Statistics 19).

## 5. Conclusions

In conclusion, we identified 51 *TaAAPs* in the wheat genome, and these genes were classified into six subfamilies based on phylogenetic and sequence analyses. The chromosome location and gene duplication analysis displayed significant gene duplication events among an extended *TaAAPs* gene family. Collinearity relationship analysis revealed several orthologous *AAPs* between wheat and other species. Moreover, cis-element analysis of promoter regions and transcriptome data suggested that the *TaAAPs* exhibited distinct expression patterns during salt stress. A *TaAAP1* gene was cloned and transformed into wheat. Overexpressing *TaAAP1* enhanced the salt tolerance by increasing the expression of ethylene synthesis genes (*TaACS6*/*TaACS7*/*TaACS8*) and accumulating more ethylene. The present study broadens our knowledge of the role of the *TaAAPs* family in the wheat genome as well as information on systematics, phylogenetics, gene duplication, and proved that overexpressing *TaAAP1* enhances salt tolerance by regulating ethylene production.

## Figures and Tables

**Figure 1 ijms-24-13800-f001:**
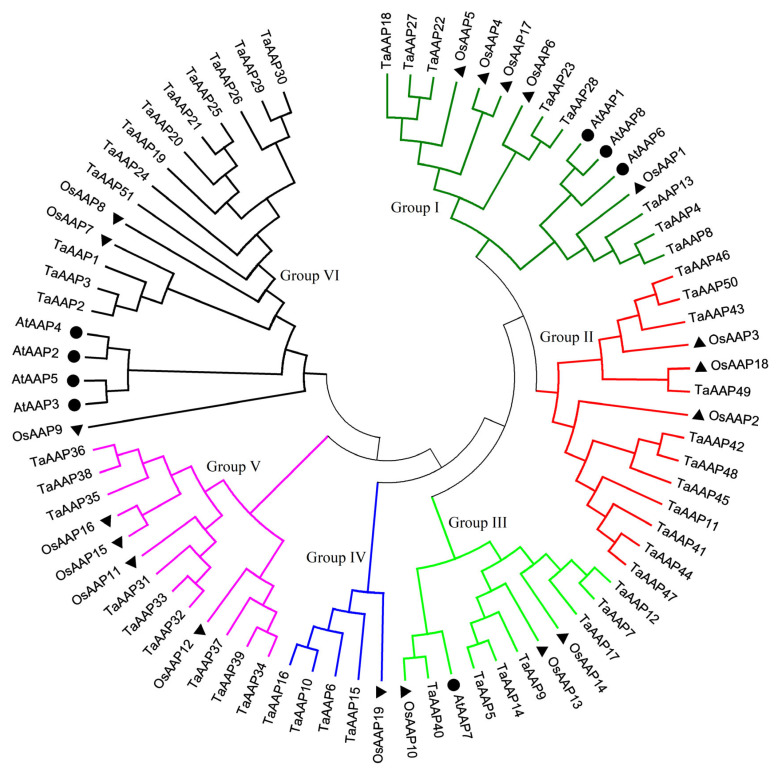
Neighbor-joining (NJ) phylogenetic tree of AAPs in wheat (*Triticum aestivum* L.), *Arabidopsis* (*Arabidopsis thaliana* L.), and rice (*Oryza sativa* L.). The tree was generated using full-length AAP protein sequences. *Arabidopsis* (black dots); rice (black triangles). We found six major phylogenetic groups, indicated with different colored lines.

**Figure 2 ijms-24-13800-f002:**
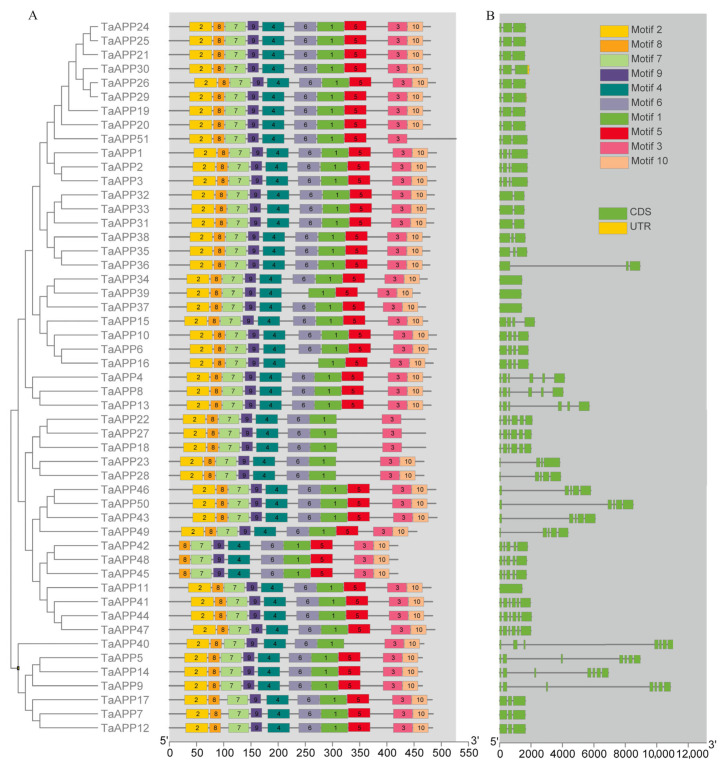
Phylogenetic relationships, conserved motifs, and gene structure in TaAAPs. (**A**) The phylogenetic tree was constructed based on full-length protein sequences using the MEGA 6.0 software with a calibration parameter of 1000. Ten conserved motifs were shown with different colored boxes. (**B**) Exon–intron structure of *TaAAP* genes, with the yellow and green boxes representing the UTR and exon sequences, respectively; the black horizontal lines represent the introns.

**Figure 3 ijms-24-13800-f003:**
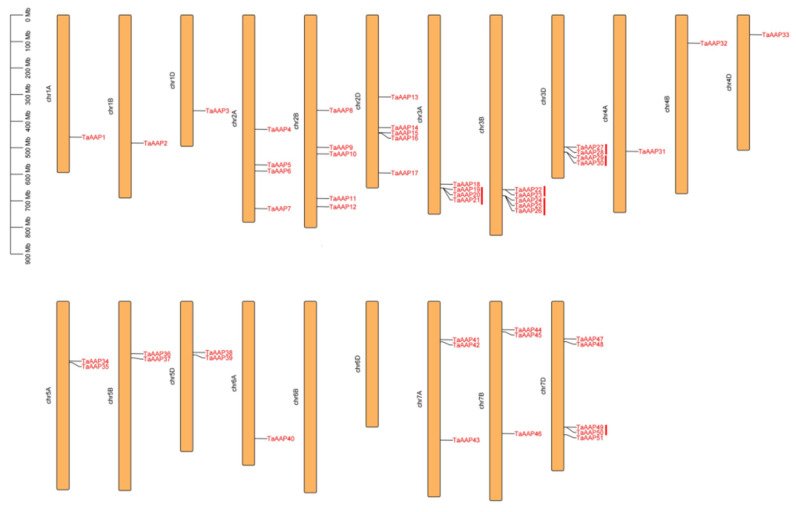
Chromosomal localization and gene duplication events of *TaAAP* genes. Red vertical lines represent tandem duplicated genes; the scale on the left is in mega bases (Mb).

**Figure 4 ijms-24-13800-f004:**
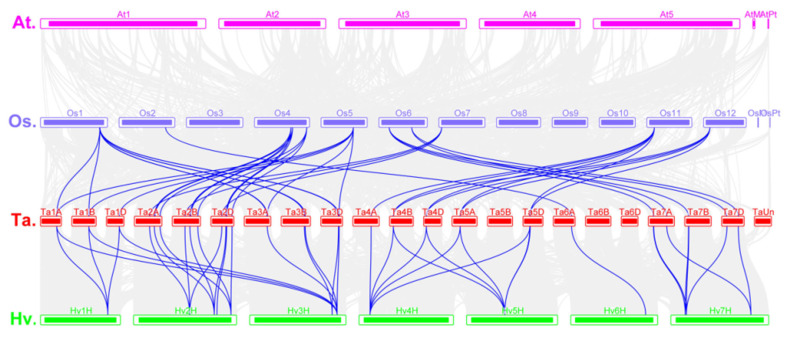
Synteny analysis of TaAAPs between wheat, *Arabidopsis*, rice, and barley. Blue lines represent the syntenic AAP gene pairs for different species. Gray lines represent collinear blocks in plant genomes. The species represent wheat, *Arabidopsis*, rice, and barley (*Hordeum vulgare*).

**Figure 5 ijms-24-13800-f005:**
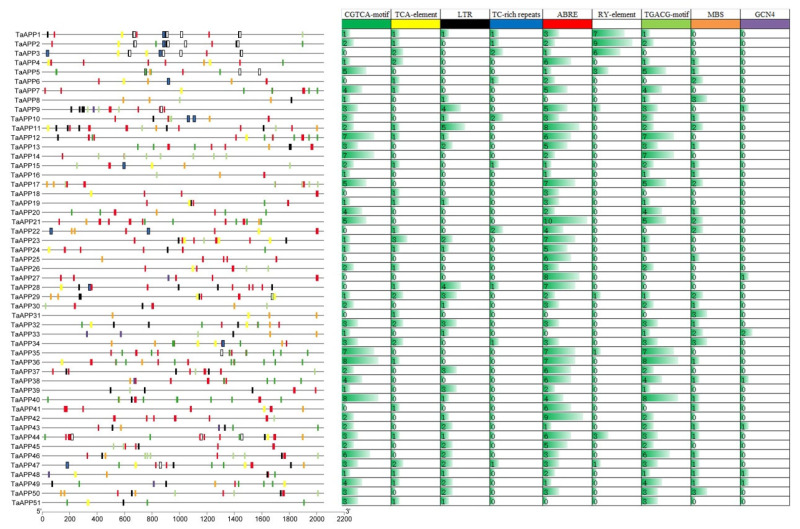
cis-element analysis of promoter sequences of *TaAAPs*. We selected a 2.0 kb region upstream of the transcription start site (TSS) and analyzed it in silico using the Plantcare website. The analysis revealed several cis-regulatory elements located in the promoters of *TaAAP* genes. Different colors and capital letters represent different cis elements. CGTCA motif: involved in methyl jasmonate (MeJA)-responsiveness; TCA element: involved in salicylic acid responsiveness; LTR: involved in low-temperature responsiveness; TC-rich repeats: involved in defense and stress responsiveness; ABRE: involved in abscisic acid responsiveness; RY element: involved in seed-specific regulation; TGACG motif: involved in MeJA-responsiveness; MBS: MYB binding site involved in drought inducibility; GCN4 motif: involved in endosperm expression.

**Figure 6 ijms-24-13800-f006:**
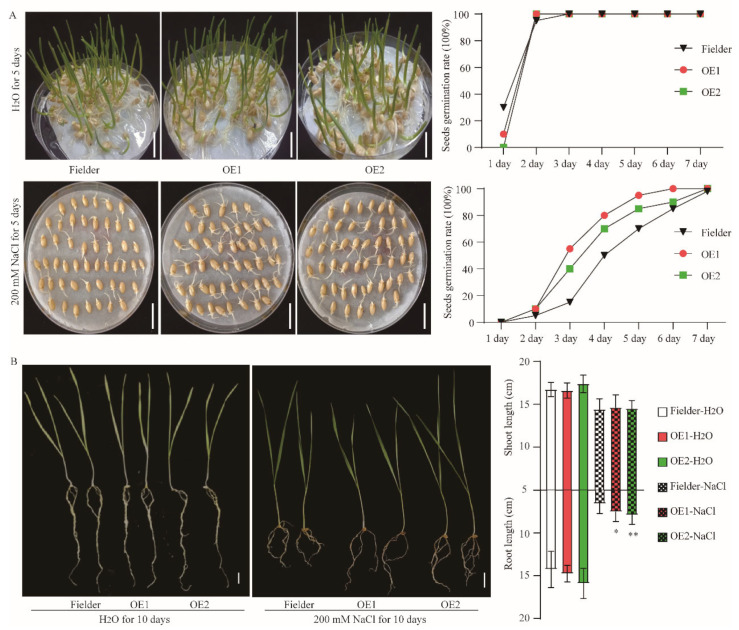
*TaAAP1* overexpressing enhanced salt tolerance in wheat. (**A**) Fifty seeds of Fielder and overexpression lines germinated in water or 200 mM NaCl condition for 7 days. The germination rate was calculated every day. Three biologicals were repeated. Bar = 2 cm. (**B**) Phenotype of 7-day-old wheat seedling in 1/2 Hoagland solution with or without 200 mM NaCl for 10 days. Shoot and root length of seedlings were calculated. The data are presented as the mean ± SD of three independent experiments. Asterisks above each column indicate statistical differences compared with Fielder in NaCl condition (* *p* < 0.05; ** *p* < 0.01). Bar = 5 cm.

**Figure 7 ijms-24-13800-f007:**
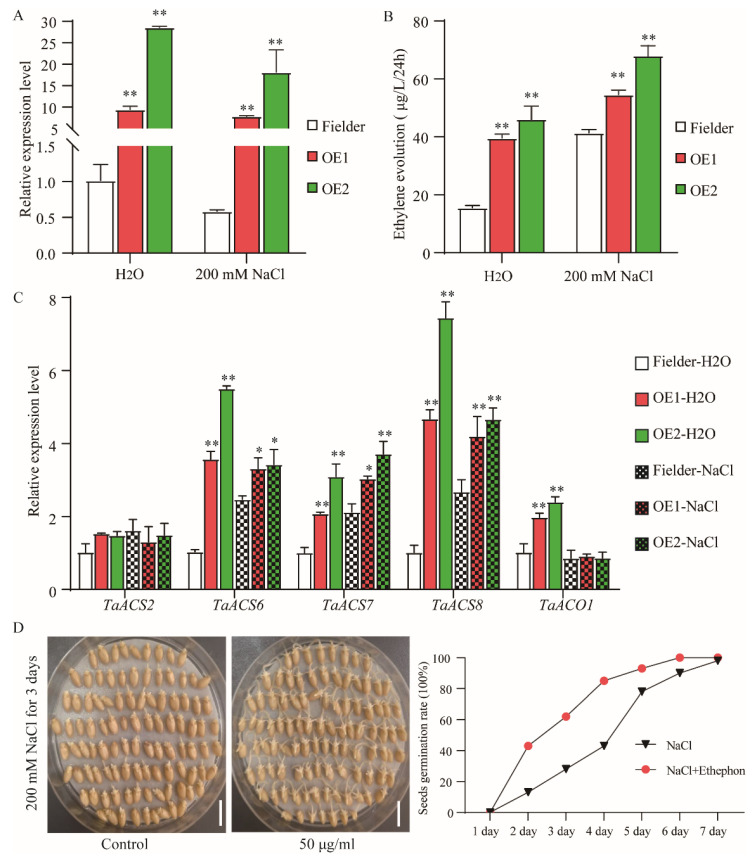
*TaAAP1* overexpressing promotes ethylene accumulation. Seven-day-old wheat seedling in 1/2 Hoagland solution with or without 200 mM NaCl for 6 h. Seedlings were harvested and the expression level of *TaAAP1* (**A**) and ethylene biosynthesis genes (**C**) were analyzed by RT-qPCR. (**B**) Ethylene production in overexpressed *TaAAP1* and Fielder lines in 1/2 Hoagland solution with or without 200 mM NaCl for 24 h. (**D**) One hundred seeds of Fielder and overexpression lines germinated in 200 mM NaCl with or without 50 μg/mL ethephon condition for 7 days. The germination rate was calculated every day. Bar = 1 cm. The data are presented as the mean ± SD of three independent experiments. Asterisks above each column indicate statistical differences to the Fielder in water (* *p* < 0.05; ** *p* < 0.01).

## Data Availability

The data and materials that support the findings of this study are available from the corresponding author upon reasonable request.

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
