# Peer review of "Genome-Wide Analysis of the Amino Acid Permeases Gene Family in Wheat and TaAAP1 Enhanced Salt Tolerance by Accumulating Ethylene"

_ijms, 2023, doi:10.3390/ijms241813800_

Round 1

Reviewer 1 Report

Comments to the manuscript

Genome-wide analysis of the amino acid permeases gene family in wheat and TaAAP1 enhanced salt-tolerance by accumulating ethylene

Authors: Kai Wang, Mingjuan Zhai, Dezhou Cui, Ran Han, Xiaolu Wang, Wenjing Xu, Guang  Qi, Xiaoxue Zeng, Yamei Zhuang and Cheng Liu

The manuscript presents the annotation of amino acid permeases (AAP) in wheat, their domain analysis, analysis of regulatory elements of TaAAP genes, assignment of AAP genes to chromosomes and demonstrates the role of TaAAP1 in salt tolerance via enhanced ethylene production. The results provide new information on the group of proteins with potential effects on such important traits as yield, seed protein content and stress tolerance.

Some corrections are recommended for the manuscript.

§  Please decipher APC in lane 45, 1-MCP in lane 147, CDS in lane 354, JA in lane 418, SA in lane 419, MeJA in lane 435

§  WheatOmics database is mentioned in the Results in lane 291 but not in Materials and Methods. Please provide all databases and tools mentioned in the Results in Materials and Methods as well.

§  In 2.6. Expression analysis of the TaAAPs gene family from RNA-Seq Data please give the details of obtaining those data or if this information has been published provide the references.

§  In Materials and methods, please describe details of production of TaAAP1-overexpressing lines (the construct, the method of transformation etc). Moreover, the cloning of TaAAP1 and the production of GM plants was mentioned in the conclusions of the manuscript: ‘A TaAAP1 gene was cloned and transformed into wheat’ (lanes 634. 635).

§  In Materials and methods, please provide the statistical methods employed for analyzing significance of differences.

§  In Fig.1 please add a triangle to OsAAP19.

§  Chromosome 4D is absent in Fig.3.

§  It seems that it should be 5 TaAAPs in lane 373 instead of 6.

§  The following references in the text of the manuscript are absent in the list of references: Sajid et al. 2018 (lane 146), Wang et al. 2019 (lanes 159, 578), Wang et al. 2021 (lane 263), Fedorov et al. 2002; Babenko et al. 2004 (lane 346), Holub et al. 2001; Chen et al. 2021 (lane 380).

§  In lanes 420, 421 there is ‘muti’- what does it mean?

§  Please reformulate the sentence in lanes 458-463 as it states that “... no difference in germination rate between transgenic and Fielder under normal condition and 200 mM NaCl, but the germination potential significant higher in overexpression lines...” It could be understood that germination did not differ between Fielder and the overexpression lines, but what is the ‘germination potential’, which is higher in the overexpression lines?

§  In the caption of Fig. 6 there is a sentence: ‘Three biological was repeated.’

§  In the caption of Fig.7, the part (C) is omitted.

§  In the caption of Fig.7, lane 509, it is written that “One hundred seeds of Fielder and overexpression lines  germinated in 200 mM NaCl with or without 50 μg/ml ethephon condition for 7 days..”. However, as described in section 2.8 in lanes 257-260, the overexpression lines were not studied with etephon, and the seed sample was 50 and not 100 (“For ethephon treatment, Fifty-seeds of Fielder were germinated...” (lane 257)). So please correct these discrepancies.

§  In discussion (lane 572) there is a phrase about ‘iron homeostaisi during salt stress’. Iron homeostasis was not described in the results.

Please check lanes 67, 97, 158, 172, 173, 250, 257, 392, 461, 476, 539, 563, 572.

Author Response

Many thanks for your valuable suggestions and comments for improving our manuscript. We have carefully revised the manuscript and thotoughly addressed your concerns. The main revisions and revised MS are detailed in the attachment.

Reviewer 2 Report

This research concerning the study of amino acid permeases genes was performed at a good level with the involvement of bioinformatics tools. A lot of careful work has been done. Valuable data on 51 wheat permease genes have been obtained, which can be of great value to both plant physiologists and wheat breeders.

However, these advantages of this work are also its disadvantages. Since there is too much bias towards bioinformatics. Most of the results are obtained with the help of software. Behind this, a living plant is lost.

My main claim to this work is the lack of data on the generation of transgenic wheat plants. And the authors draw some conclusions based on transgenes. In the Materials and Methods section, there is no methodology for these procedures. How was TaAAP1 cloned? In what genetic construct is it placed? What variety of wheat was used for genetic transformation? What kind of transformation was used? How were transgenic plants obtained? How was their transgenic nature confirmed? I draw your attention to the fact that wheat is not a crop that can be easily transformed. However, for the authors, it easily appears from somewhere ... In the Results section, this is not there either .... Authors should either remove all references to transgenic plants and expression from the manuscript and move them to another article, or answer the questions posed.

Other remarks. Lines 24-25 “cis-element analysis of promoter regions and transcriptome data suggested that the TaAAPs response to salt stress” – the authors did not conduct experiments, so it should be written: they can respond to stress.

Lines 26-27 The manuscript does not describe the production of transgenic wheat. Therefore, such a statement in the Abstract looks doubtful.

The Introduction section is very large. It could be shortened

References to sources in the text must be made in accordance with the requirements of MDPI.

Author Response

(The authors gave the same response as above.)

Round 2

Reviewer 2 Report

The authors have carefully worked on improving the manuscript. Although I believe that the process of obtaining transgenic wheat plants is not well described, I hope that in their future publications the authors will pay more attention to this issue.

I believe that the manuscript may be accepted for publication in its present form.